

# No apparent correlation between honey bee forager gut microbiota and honey production

Melissa A. Horton[1], Randy Oliver[2] and Irene L. Newton[1]

[1] Department of Biology, Indiana University, Bloomington, IN, United States
[2] ScientificBeekeeping.com, Grass Valley, CA, USA

## ABSTRACT

One of the best indicators of colony health for the European honey bee (*Apis mellifera*) is its performance in the production of honey. Recent research into the microbial communities naturally populating the bee gut raise the question as to whether there is a correlation between microbial community structure and colony productivity. In this work, we used 16S rRNA amplicon sequencing to explore the microbial composition associated with forager bees from honey bee colonies producing large amounts of surplus honey (productive) and compared them to colonies producing less (unproductive). As supported by previous work, the honey bee microbiome was found to be dominated by three major phyla: the Proteobacteria, Bacilli and Actinobacteria, within which we found a total of 23 different bacterial genera, including known "core" honey bee microbiome members. Using discriminant function analysis and correlation-based network analysis, we identified highly abundant members (such as *Frischella* and *Gilliamella*) as important in shaping the bacterial community; libraries from colonies with high quantities of these *Orbaceae* members were also likely to contain fewer *Bifidobacteria* and *Lactobacillus* species (such as Firm-4). However, co-culture assays, using isolates from these major clades, were unable to confirm any antagonistic interaction between *Gilliamella* and honey bee gut bacteria. Our results suggest that honey bee colony productivity is associated with increased bacterial diversity, although this mechanism behind this correlation has yet to be determined. Our results also suggest researchers should not base inferences of bacterial interactions solely on correlations found using sequencing. Instead, we suggest that depth of sequencing and library size can dramatically influence *statistically significant* results from sequence analysis of amplicons and should be cautiously interpreted.

# INTRODUCTION

As with other livestock producers, there is interest in the beekeeping industry as to whether it would be of benefit to feed probiotic bacteria to honey bee colonies. Any potential benefit as far as productivity of the colony would presumably be the result of the colonization of the bee guts by the introduced probiotic. By extension then, and in order to select specific bacteria as probiotics, there should be evidence that a difference

Corresponding author
Irene L. Newton,
irnewton@indiana.edu

exists between the gut microbiota of productive (those that produce a surplus of honey) and non productive colonies in the field. In this study, we tested the hypothesis that the difference in productivity of colonies in the field was associated with a difference in the community structures of their respective gut microbiota. These bacterial communities have been best characterized in worker bees, are consistently found across geography and season (*Martinson et al., 2011*), and are thought to consist of a core group of bacterial clades some of which have genus and species designations (*Kwong & Moran, 2012*; *Engel, Kwong & Moran, 2013*). Three major bacterial phyla dominate the honey bee microbiome and include the Proteobacteria (including *Gilliamella, Frischella*, and *Parasaccharibacter*) the Firmicutes (e.g., *Lactobacillus sp.* Firm-4 and Firm-5), and the Actinobacteria (such as *Bifidobacterium*). This microbial community is thought to be socially transmitted within the hive via interaction with hive components and fecal material of congeners (*Martinson, Moy & Moran, 2012*; *Powell et al., 2014*). These communities are surprisingly consistent between workers and their increased prevalence during honey bee development would seem to suggest that these bacteria may contribute to the health of the bee. Some potential functions for these groups with regards to honey bee health and nutrition have been explored using meta-omic methods (*Engel, Martinson & Moran, 2012*; *Lee et al., 2015*). These potential metabolic contributions include the expression of enzymes used to degrade complex polysaccharides in the honey bee diet for which the host does not produce the appropriate enzymes or pathways (*Engel, Martinson & Moran, 2012*; *Lee et al., 2015*). Additionally, researchers have explored the ability of some of these bacterial strains to competitively exclude pathogens *in vitro* (*Vojvodic, Rehan & Anderson, 2013*; *Corby-Harris et al., 2014*), suggesting that the colonization of the honey bee by the microbiome may help to prevent disease (*Olofsson & Vasquez, 2008*; *Vasquez & Olofsson, 2009*; *Forsgren et al., 2010*; *Vasquez et al., 2012*). It is therefore possible that the composition of the bacterial community in the honey bee may influence health and productivity.

Here we explore microbial correlates in honey bees from productive and unproductive colonies. We utilize 16S rRNA gene amplicon sequencing to characterize bacteria associated with foraging bees sampled from the most or least productive colonies in each of 7 different apiaries. We sampled foraging bees as this worker caste collects nectar that is made into honey and the microbial composition in their guts could impact productivity. We identified weak trends in bacterial community composition correlated with colony productivity including an increase in diversity in productive colonies and an increase in the presence of one specific *Lactobacillus* species (Firm-4). These trends were associated with a decrease in prevalence of one dominant bacterial group, the $\gamma$-proteobacterial genus *Gilliamella*. We also show that although *Gilliamella* strains in culture do not competitively inhibit the growth of honey bee specific bacteria, their dominance in the dataset correlated with a decrease in diversity and a greater ability to detect rare community members. Our results point to *potential* microbial signatures of honey bee productivity that deserve further exploration.

## METHODS

### Honey bee colony management and productivity measurements

All samples were collected from apiaries ($N = 7$) in the Sierra Foothills (California) in nonagricultural areas (minimal pesticide exposure) consisting of mixed forest and grasslands in the Foothill (Blue Oak) to Lower Montane (Yellow Pine) belts. Each apiary contained approximately 24 hives. At the end of the local honey flow (early August 2012), we determined by inspection the 2 most productive hives in each apiary, and the 2 least productive, as judged by the amount of surplus honey stored. We excluded from the "least productive" group any colonies that were weak (fewer number of workers) or showed signs of disease, so that we would be comparing only those colonies of roughly equal strength and health.

In order to standardize the bees sampled, we collected returning foragers, as indicated by their carrying of pollen loads. Such sampling would standardize the age of the bees sampled, as well as their diet, since forager bees no longer consume pollen, and subsist upon a diet of nectar, honey, and jelly begged from younger bees. We collected returning pollen foragers from the entrances, humanely dispatched them by pinching off their heads, and carefully extracted their guts in the field with alcohol-sterilized forceps by clamping the tip of the last abdominal segment and slowly pulling the entire gut (hindgut, midgut, and usually the honey stomach) from their bodies. The pooled guts for each hive (5 total) were immediately dropped into a vial of RNA later (Life Technologies, Carlsbad, California, USA) for preservation, put on ice, frozen within 2 h, and shipped frozen for processing at Indiana University, Bloomington.

### Honey bee sampling, extraction of nucleic acids and library generation for Illumina

Upon receipt of the dissected honey bee samples, RNA-later was decanted from each vial and subsequently each colony sample was processed using the MoBio PowerSoil DNA extraction kit. DNA quantities from each colony sample were measured spectrophotometrically and normalized before use in each of three polymerase chain reactions using Earth Microbiome barcoded primers (515F and 806R, tags rcbc1-42). The following modifications were made to the Earth Microbiome amplification protocols: HF Phusion master mix (New England Biolabs, Ipswich, Massachusetts, USA) was used in combination with 100 ng of template DNA with 25 ul volume reactions. Two reactions were completed for each template and pooled after amplification. After reactions completed, amplicons were visualized on an agarose gel then cleaned using a PCR cleanup kit (Qiagen, Venlo, Netherlands) before normalization and pooling for sequencing. Libraries made in this fashion were sequenced on an Illumina MiSeq using 300 cycles (SE).

### Bioinformatic analysis of data

All bioinformatics processing was performed in the Mothur microbial ecology suite (*Schloss et al., 2009*). Individual bins containing sequences from each colony were identified using the 10bp index used in the amplification of that sample (no ambiguous base

pairs allowed). The sequencing output was then screened for basic quality (maxambig = 0, maxlength = 300, qaverage = 21) before alignment to the silva reference (silva.v4.fasta) and sequence alignment trimming to homologous regions. Sequences were preclustered and checked for chimeras using uchime as implemented in Mothur. Resulting sequences were classified at a confidence threshold of 80 using the RDPII-NBC with the honey bee specific training set + SILVA (*Newton & Roeselers, 2012*). Data are available to reviewers upon request and will be deposited in the NCBI SRA upon acceptance of the manuscript.

## Statistical and network tests

All statistical analyses were performed using the SPSS software suite (v21). For Discriminant Function Analysis (DA), Kruskall-Wallace and Spearman's Rho correlations, libraries from each colony were subsampled to 10,000 using in-house scripts. DA allows us to identify patterns in pre-determined clusters of samples (in this case, productive and unproductive colonies) and we used Wilks's lambda ($p < 0.05$) to determine significance. In addition, we were able to cross-validate our model generated by the DA to sort samples into their respective bins (productive versus unproductive) with high confidence. The Kruskall-Wallace and discriminant function analyses were used on a single subsampling in order to identify bacterial families, species, or genera that correlated with colony productivity. The Spearman's Rho correlations were used on the same subsampled set in order to identify correlations between different bacterial members in the community. In order to determine 95% confidence intervals for presence of different bacterial families between productive and non-productive colonies, the entire dataset was subsampled, generating *in silico* libraries of size 10,000 a total of 1,000 times. Distributions of counts for important bacterial families were compared and if 95% confidence intervals did not overlap, then the presence of these families was considered to be statistically significant with regards to colony productivity. Statistically significant ($p < 0.05$) Spearman's Rho correlations between different bacterial species were visualized in cytoscape. Raw correlations were parsed with in house scripts and this file (in tab delimited format) was used as input to Cytoscape (v. 3.0.2). Edges in the network were sized (larger = extent of correlation ($R^2$) and the direction of the correlation (negative (red) or positive (blue)) was colored.

## Co-culture and microbiological protocols

The Newton Laboratory honey bee bacterial strain bank was used as a source of bacterial isolates. In brief, bacteria from the honey bee gut and bee bread were cultured on either MRS, LB, BHI or TSA agar (at 37 °C for 48 h under anaerobic conditions) and individual colonies were isolated using a robotic colony picker (QPExpression; Genetix, Hampshire, UK). Each isolate was classified based on 16S rRNA gene sequencing and classification using the Naïve Bayesian Classifier and the honey bee specific training set (*Newton & Roeselers, 2012*). For this study, we chose a variety of isolates from the Newton Laboratory honey bee bacterial strain bank including bacteria classified as Gamma-1, Enterobacteriaceae, Bifidobacteriaceae, Firm-4, *Paenibacillus*, Bacillus, and *Fructobacillus*; 16S rRNA gene sequences are available in GenBank (KT598279, KT598280, KT598281, KT598282, KT598283, KT598284, KT598285 and KT598286). Each isolate was cultured

for 48 h in BHI broth at 37 °C under anaerobic conditions. After 48 h, culture OD600 measurements were taken and each was normalized to the lowest optical density. The bacteria were cultured alone or in co-culture in triplicate experimental replicates in all pairwise combinations. Optical densities were measured every 24 h for 3 days. To analyze the co-culture data, the expected optical density was calculated based on each isolates growth alone on day 3. If isolates grew better in co-culture, the expected optical density would be significantly above (outside of the standard deviation) of the calculated expected OD. Likewise, if one of the isolates inhibited the growth of the other, the OD would be below the expected value. Optical density differences between the expected and actual ODs above or below the standard deviation were considered significant. Data were analyzed using Excel. Evolutionary analyses were conducted in MEGA6 using the Maximum Likelihood method based on the General Time Reversible Model with a gamma distribution, invariable sites, and 100 bootstrap replicates (*Tamura et al., 2013*).

## RESULTS

### Foraging bees from both productive and unproductive colonies host many bacterial genera

Illumina sequencing produced a large number of reads from the sampled honey bee colonies (3,252,621 raw reads and 2,903,512 post filtering), allowing a deep sampling of the honey bee microbial community (Fig. 1). Sequences matching important and recurring clades considered part of the honey bee core microbiome (*Martinson et al., 2011*) were found at high frequency within the dataset (Table 1). Specifically, the dataset from each colony was dominated by sequences homologous to the Orbaceae genera *Gilliamella* and *Frischella* (Table 1). Other core honey bee microbiome members were also identified across all colonies and included *Snodgrassella* species, firm-5, firm-4, alpha-2.1 and alpha-2.2 (Fig. 1). In addition to the core honey bee microbiome, sequences in the dataset matched other bacterial members, and appeared across all sampled colonies, and at >1% frequency in the total dataset. These organisms included several enteric genera (*Acinetobacter, Salmonella, Pantoea*, and *Pseudomonas*), Bacilli (*Staphylococcus, Lactobacillus*) and $\alpha$-proteobacteria (*Ochrobactum, Saccharibacter*).

### Forager microbial communities do not appear to correlate with colony productivity

The dataset produced by these samplings can be analyzed in two fundamentally different ways. The membership within each colony can be kept static and subsampling within colonies can be used to assess differences between productive and unproductive colonies. Alternatively, colony microbiome membership can be aggregated, based on productivity to create *in silico* libraries through statistical resampling of the two pools (productive vs. unproductive). The first approach retains noise resulting from between colony variability, requires us to sample at or below the smallest library size (25,000) but allows us the power of biological replicates ($N = 14$). The second approach averages across colonies, tempering the effects of inter-colony variability in microbiome composition but preserving the single variable (colony productivity), and allowing for large statistical resamplings (in this case,

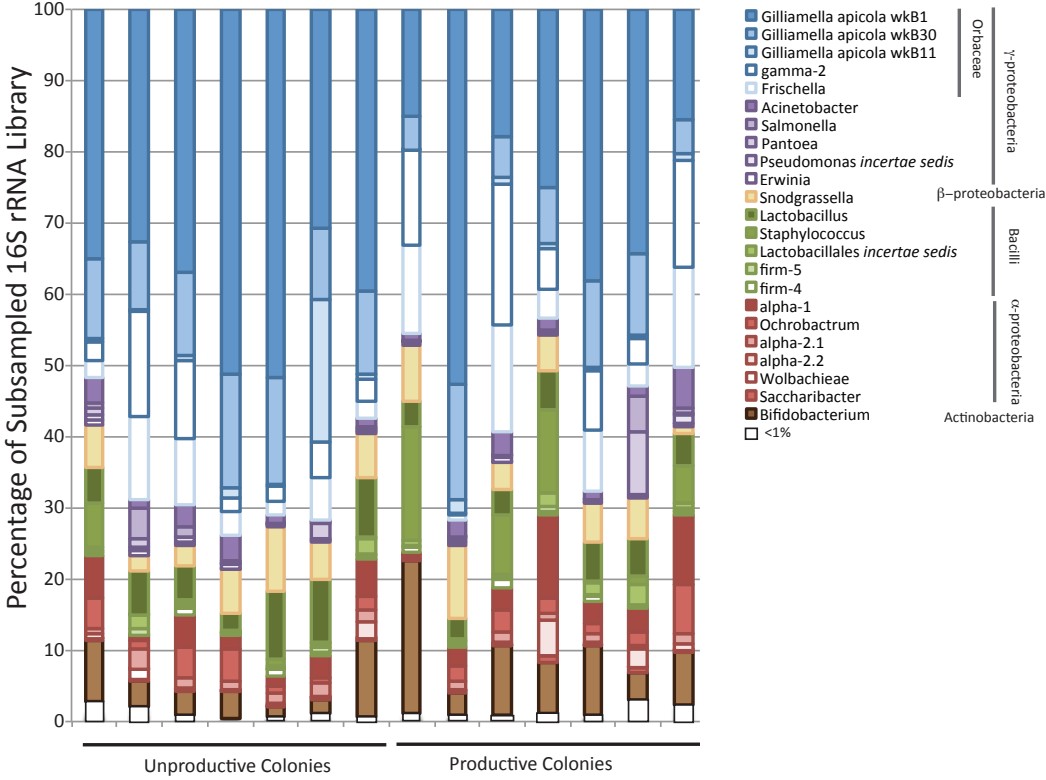

**Figure 1** **Bar chart showing diversity of bacteria within each sampled forager population.** *Colony productivity does not correlate with forager gut microbiomes.* The relative abundance of different bacterial families in productive and non-productive colonies as classified by the RDPII-NBC using a honey bee specific training set combined with the SILVA taxonomy. Classifications are colored and sorted based on phylum level membership. Colonies were classified as productive or non-productive based on gross metrics (see methods).

we generated 100 libraries from each condition, each of size 10,000). Below, we present the results of these two different approaches.

Both Unifrac (weighted and unweighted) and PCA analyses (as implemented in Mothur) were utilized to determine if foragers from productive colonies differed in their microbial profiles compared to those from unproductive colonies. Neither test produced significant clustering of these colony types ($p > 0.05$ in each case). To test the hypothesis that colony productivity contributed to a distinguishing microbiome profile, we performed a step-wise discriminant function analysis (DA) on rarefied, subsampled sequence classifications (at the genus level) specific to each biological replicate. Microbiome composition could readily discriminate productive and unproductive colonies (Wilks's lambda coefficient = 0.204; $\chi^2 = 17.483$; df = 4; $p = 0.002$). We further probed which canonical discriminant functions (in this case, bacterial taxonomic groups) were primarily contributing to the difference between these colony types. We discovered four significant predictors of colony productivity: the most significant predictor was *Bacilli incertae sedis* (Wilks's lambda = 0.746; $F = 4.437$; $p = 0.055$), while the next most significant predictors were the *Comamonodaceae incertae sedis* (Wilks's lambda = 0.465; $F = 6.904$; $p = 0.010$),

**Table 1 Abundance of different phylotypes found in the entire dataset.** Average number of reads per colony classified as honey bee core microbiome members, using the RDPII-NBC and a honey bee specific training set combined with the SILVA taxonomy. For each bacterial group, minimum and maximum number of reads also listed.

| Bacterial group | Average per colony (Num Reads) | Min (Num Reads) | Max (Num Reads) |
|---|---|---|---|
| Bifidobacterium | 722.6 | 1 | 2,238 |
| Frischella | 637.7 | 48 | 1,466 |
| Gilliamella_apicola_wkB1 | 3,200.9 | 1 | 5,231 |
| Gilliamella_apicola_wkB11 | 439.6 | 17 | 3,376 |
| Gilliamella_apicola_wkB30 | 1,001.8 | 37 | 1,684 |
| Snodgrassella | 573.7 | 108 | 1,059 |
| alpha-1 | 438.0 | 22 | 1,154 |
| alpha-2.1 | 146.4 | 5 | 533 |
| alpha-2.2 | 112.2 | 15 | 493 |
| beta | 2.1 | 0 | 6 |
| firm-5 | 29.8 | 9 | 74 |
| firm-4 | 92.9 | 9 | 502 |
| gamma-2 | 711.3 | 50 | 1,999 |

followed by *Enterobacteriales incertae sedis* (Wilks's lambda $= 0.305$; $F = 8.340$; $p = 0.004$), and alpha-1 (Wilks's lambda $= 0.204$; $F = 9.752$; $p = 0.002$). Because a DA analysis, although robust, can be affected by violations of assumptions (such as multicollinearity), we also explored differences between productive and unproductive colonies using the Kruskall Wallis test (a non-parametric ANOVA equivalent). In this case, we are testing if our microbiome libraries from productive and unproductive colonies are similar enough to each other to appear as though they came from the same source. *Bacilli incertae sedis* (Kruskal Wallis, $\chi^2 = 4.292$; df $= 1$; $p = 0.038$), and *Comamonadaceae incertae sedis* (Kruskal Wallis, $\chi^2 = 4.885$; df $= 1$; $p = 0.027$) were identified as significantly different between productive and unproductive colonies. However, both of these bacterial groups (*Bacilli incertae sedis* and *Comamonadaceae incertae sedis*) occur at extremely low frequency in our dataset (mean $<1$ across all colonies with max $= 5$). It is very unlikely that these organisms are contributing to significant physiological differences between colonies. However, the presence of these organisms in our libraries could be an indicator of colony microbiome diversity or evenness. We calculated Shannon's diversity index (H) and species evenness (H/ln(S)) for each colony based on classifications of genera. The median values for diversity and evenness for productive colonies (H $= 2.41$; H/ln(S) $= 0.69$) were higher than that for unproductive colonies (H $= 2.19$; H/ln(S) $= 0.64$). Although richness values among colony types trended towards higher values for productive colonies, this difference was not statistically significant (Kruskal Wallis, $\chi^2 = 3.188$; df $= 1$; $p = 0.074$).

The second major way in which we mined the dataset for differences based on colony productivity relied on *in silico* resampling, generating 100 libraries of size 10,000 from pooled sequences from productive or unproductive colonies. The frequency distribution for different taxa resulting from these resamplings were plotted and 95% confidence

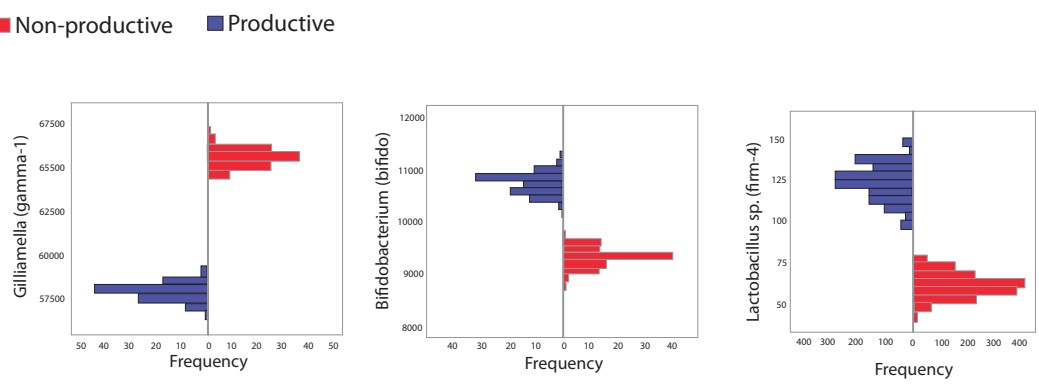

**Figure 2** **Histogram showing frequency of bacteria in subsampled in silico libraries.** Histograms showing the frequency of three different, bee-specific bacterial groups (gamma-1, bifido, and firm-4) found within *in silico* replicated samples from productive and unproductive colonies. Pooled data from productive (blue) and unproductive (red) colonies were used and resampling, without replacement, was performed a total of 100 times, each time creating an *in silico* replicated library of 10,000 sequences. All comparisons between unproductive and productive colonies are statistically significant ($p < 0.05$); 95% confidence intervals did not overlap between colony types.

intervals calculated. Sequences matching the *Bifidobacteriaceae* family were identified as more abundant in productive versus unproductive *in silico* replicates. However, these trends were not found in *biological* replicates (maximum, minimum counts and SD in *biological* replicates were: productive = 1,520, 434; SD = 388; unproductive = 2,942, 186; SD = 918). Other trends supported by both the *in silico* and the *biological* replicates included the reduction of the *Orbaceae* family in productive colonies (maximum, minimum counts and SD in *biological* replicates were: productive = 7,075, 4,159; SD = 943; unproductive = 7,335, 4,079; SD = 1,203) and an increase in Firm-4 in productive colonies (maximum, minimum counts and SD in *biological* replicates were: productive = 425, 15; SD = 132; unproductive = 110, 2; SD = 35).

## The presence of certain bacterial families precludes the identification of others

Most environmental sequence analyses methods are based on sampling of a *pool* of sequences. The diversity and evenness of the environment will affect the ability of the researcher to effectively characterize the community; if diversity is high but evenness low, the preponderance of a few taxa will preclude the identification of others. For example, our finding that the presence of rare members (such as *Comamonodaceae incertae sedis* and *Enterobacteriales incertae sedis*) allows one to discriminate between productive and unproductive colonies suggests that other, more highly abundant bacteria, may be correlated with a reduction other bacterial members. We therefore investigated potential correlations in the dataset between the presence and absence of bacterial sequences in our biological replicates (classified at the family level). Spearman's Rho correlations were calculated based on bacterial family counts across all 14 honey bee colony samples. One striking result was the strong negative correlations between the presence of sequences matching the *Orbaceae* (of which *Gilliamella* and *Frischella* are members) and the presence

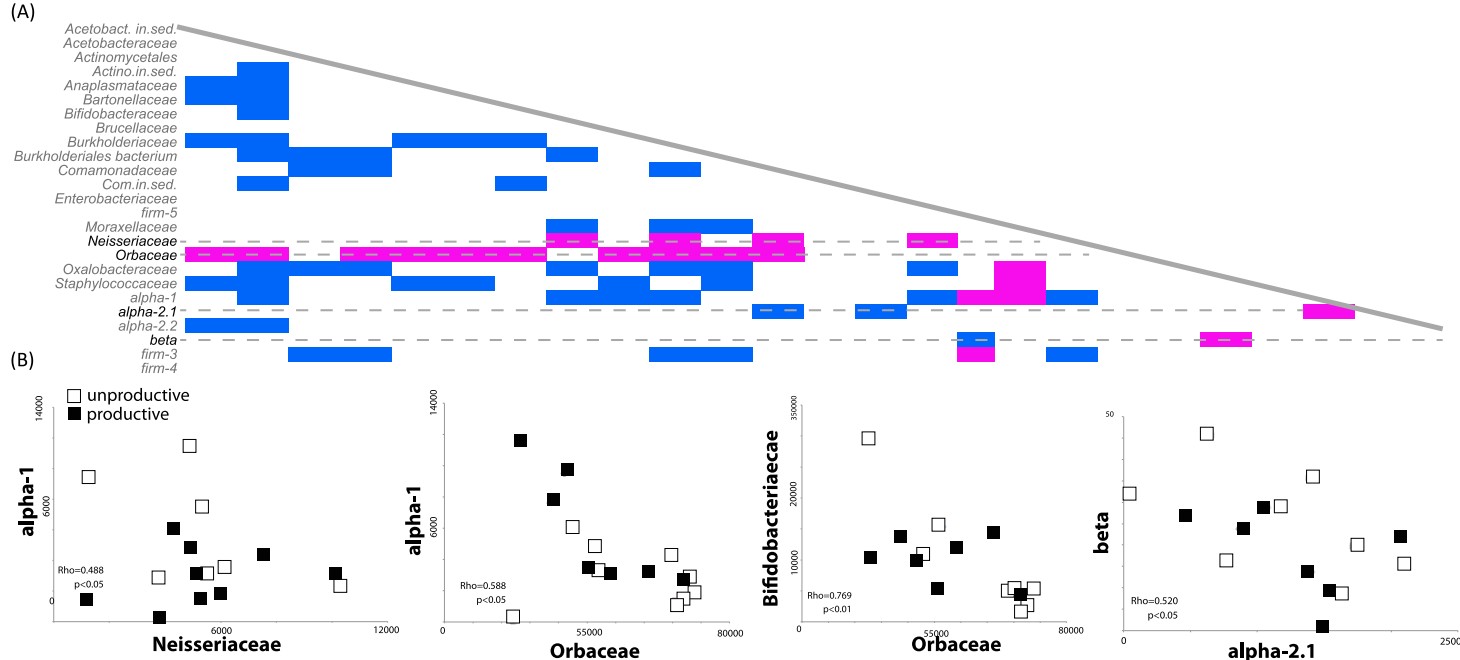

**Figure 3  Correlated abundance profiles for Orbaceae and other bee specific groups.** Orbaceae abundance negatively correlates with presence of many bacterial groups in the honey bee gut. (A) The presence and absence of different bacterial families was investigated using Spearman's Rho correlations in rarefied, biological replicates. Positive correlations are indicated in blue while negative correlations are highlighted in pink. Across all of the biological replicates ($N = 14$), the presence of sequences from the family Orbaceae (which includes Gilliamella and Frischella) negatively correlated with the abundance of many other families. Other negative correlations were detected between the Neisseriaceae and alpha-1 as well as between beta and alpha-2.1. (B) Demonstrative linear correlations based on counts in each honey bee colony sample used to generate part (A).

of many other bacterial members. A total of 14 different bacterial families were negatively correlated with the presence of *Orbaceae* (Fig. 3A), including families thought to be important to honey bee health, such as *Bifidobacteriaceae* and Alpha-1 (Fig. 3B).

## Co-culture assays between honey bee bacterial members do not support co-occurrence data

Our co-occurrence data suggested that certain bacterial members (specifically *Gilliamella apicola* within the *Orbaceae*) are interacting negatively with other bacteria within the honey bee gut and that these members may be suppressing the growth of other bacteria. We sought to test this hypothesis through experimental validation via co-culture. We cultured isolates from the honey bee in BHI medium (Table 2) and normalized optical density measures before combining the cultures, in triplicate, and also growing them in isolation as a control. These cultures were incubated for 72 h before optical densities were again measured. We calculated an expected optical density based on the growth of each strain in isolation, during that same time period. For example, if the bacterial strains do not interact with each other, we would expect that their growth in co-culture would resemble their growth in isolation while if one isolate negatively suppressed the growth of another, we would expect to find decreased growth, compared to the reference. When honey bee specific bacteria are cultured in the presence of *Orbaceae* we do not find evidence of

**Table 2 Taxonomic classification of strains used in *in vitro* assays.** Taxonomic classification and top blast hit of strains used in this study representative of the honey bee associated microbial community. For each strain, the percent identity to OTUs identified in this study is also shown.

| Strain designation | Taxonomic classification | Top blast hit (Accession) | % ID |
| --- | --- | --- | --- |
| **Gamma-1_LBTET_A11** | *Gilliamella* | KF600145.1 | 97% |
| **Gamma-1_LBTET_D01** | *Gilliamella* | JQ581985.1 | 98% |
| **Enterobacteriaceae_LBTET_C07** | *Enterobacteriaceae* | KF600145.1 | 100% |
| **Paenibacillus_CIK** | *Paenibacillus* | HM566718.1 | 100% |
| **Bacillus_CII** | *Bacillus* | KF791523.1 | 98% |
| **Fructobacillus_F2** | *Fructobacillus* | KJ424425.1 | 98% |
| **Firm-4_SF6D** | Firm-4 | HM112052.1 | 99% |
| **Bifidobacteriaceae_G10-2** | *Bifidobacteriaceae* | HM534845.1 | 100% |

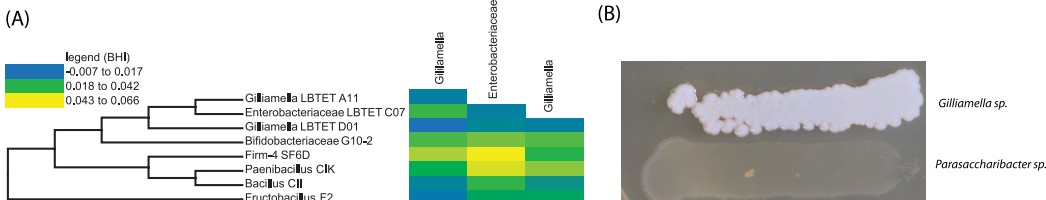

**Figure 4 *In vitro* growth assays using bee specific microbes.** Negative interactions inferred by correlations between abundance profile for honey bee specific bacteria are not confirmed by *in vitro* assays. (A) Phylogeny of bacterial isolates used in a co-culture interaction assay and co-culture assay results (heatmap). Change in optical density from expected values based on growth in isolation plotted as a heat map (yellow = more growth than expected; blue = less growth than expected). No statistically significant growth differences found for growth with *Gilliamella* compared to growth alone. (B) Growth of *Gilliamella* on hard agar plates next to *Parasaccharibacter*, another honey bee associated strain. When grown in broth co-culture, or on hard agar, *Gilliamella* species do not exhibit antagonistic interactions.

a negative interaction. In fact, two Gamma-1 isolates interacted positively with other honey bee isolates (Gamma-1 LBtet A11 with Firm-4 SF6D and Bifidobacteriaceae G10-2; Gamma-1 LBtet D01 with *Paenibacillus* CIK and Bifidobacteriaceae G10-2 (Fig. 4)). Under these co-culture conditions, it does not appear that the *Orbaceae* are suppressing the growth of honey bee specific bacteria.

# DISCUSSION

## Co-occurrence data should be cautiously interpreted

Many researchers use co-occurrence patterns in 16S rRNA gene sequencing data to hypothesize non-random interactions between microbes (*Freilich et al., 2010*; *Beman, Steele & Fuhrman, 2011*; *Steele et al., 2011*; *Faust & Raes, 2012*; *Faust et al., 2012*; *Barberan et al., 2014*); few studies, however, have then utilized *in vitro* or *in vivo* experimentation to test these hypotheses. One result of this study was the statistically significant negative correlation seen between the dominance of *Gilliamella* species and the reduction in prevalence of other bacterial members. Across all samples, our data consistently indicated that high prevalence of *Gilliamella* was associated with a reduction in bacterial taxa in the

honey bee microbiome, including the core clades of *Bifidobacterium* and Alpha-1. These results led us to hypothesize that the presence of *Gilliamella* would actively exclude the growth or activity of other honey bee associated bacteria. However, *in vitro* co-culture experiments using representative isolates could not identify an antagonistic interaction between *Gilliamella* strains and other honey bee microbes. Of course, it is possible that the specific strains isolated and utilized in the co-culture work do not *functionally* represent those from the co-occurrence data. Additionally, pairwise interactions in media certainly do not mimic the honey bee gut environment, which is multi-species and more nutritionally complex. We suggest that co-occurrence data, based on presence and absence of bacterial 16S rRNA gene sequences, should be cautiously interpreted.

## Microbial correlates with honey bee colony productivity

The honey bee microbiome has only recently been characterized using culture-independent approaches (*Cox-Foster et al., 2007*; *Engel, Martinson & Moran, 2012*; *Martinson, Moy & Moran, 2012*; *Moran et al., 2012*; *Powell et al., 2014*). Predicted functions of microbial members based on genomic (*Engel, Martinson & Moran, 2012*) or transcriptomic (*Lee et al., 2015*) data as well as *in vitro* characterizations of the metabolisms and physiology of these bacterial associates are beginning to provide hypotheses as to their contributions to host nutrition and health. As of yet, however, specific impacts of the microbiome on honey bee health or productivity have not been elucidated. Here we sought to identify microbial correlates with honey bee productivity using 16S rRNA gene amplicons from DNA isolated from forager digestive tracts. Other studies focusing on the gut microbiota of foragers have found this caste to harbor diverse microbiota (*Corby-Harris, Maes & Anderson, 2014*; *Hroncova et al., 2015*; *Kapheim et al., 2015*). Indeed, some authors have suggested that foragers may introduce microbial diversity to the colony from the natural environment (*Kapheim et al., 2015*). Overall, our data suggest that colony productivity was not consistently correlated with the forager gut microbiomes in the colonies that we sampled, although we did observe various trends, which leave open the possibility that our relatively small sample sizes may have overlooked an actual effect.

For example, we found that productive colonies trended towards increased diversity and prevalence of *Lactobacillus* species (such as Firm-4). We also found that productive colonies were less likely to be dominated by *Gilliamella* species (Fig. 2) and that the presence of these $\gamma$-proteobacteria was negatively correlated with the ability to detect other bacterial groups (Fig. 3). However, it does not appear that *Gilliamella* antagonizes other honey bee microbiome members *in vitro*, therefore the potential mechanism behind this statistical correlation has yet to be determined.

We specifically focused on one behavioral group of honey bee for this study: foragers. The logic was that the gut microbiome of these bees was representative of the cohorts of bees involved in nectar gathering and honey production. However, we did not sample the bacterial compositions of the crops of these bees. Nectar that is collected in the crop could come into contact with bacteria there, and these organisms could presumably have an effect on colony productivity. However, as the bacterial community in the crop is largely

made up of more transient, environmental organisms (such as *Lactobacillus kunkeii*) (*Corby-Harris, Maes & Anderson, 2014*), any contribution to productivity by honey bee core bacteria in the crop is unlikely. Additionally, we utilized DNA to characterize the microbial community present in the forager bee digestive tracts but did not analyze the *active* microbial community using RNA. Finally, we did not sample the other microbial environments found on the bees or associated with the colony. It is possible that these microbiomes would differ more significantly between colony types (productive vs. non-productive). All in all, our results do not appear to indicate that the feeding of certain probiotic bacteria to honey bee foragers is likely to result in increased honey production by honey bee colonies.

## ACKNOWLEDGEMENTS

We thank Stephanie Rosa and Zach Rokop for assistance in sample processing. This work was funded by Scientific Beekeeping and start up funds from Indiana University to ILGN.

### Funding

Research was funded by a grant from Scientific Beekeeping and start up funds from Indiana University to ILGN. Scientific Beekeeping is a non-profit website, supported by readers and focused on the practice of bee keeping.

### Grant Disclosures

The following grant information was disclosed by the authors:
Scientific Beekeeping.
Indiana University.

### Competing Interests

Randy Oliver is the head of Scientific Beekeeping, the organization that funded the work. Irene Newton is an Academic Editor for PeerJ.

### Author Contributions

- Melissa A. Horton performed the experiments, analyzed the data, contributed reagents/materials/analysis tools, wrote the paper, prepared figures and/or tables, reviewed drafts of the paper.
- Randy Oliver conceived and designed the experiments, performed the experiments, contributed reagents/materials/analysis tools, reviewed drafts of the paper.
- Irene L. Newton conceived and designed the experiments, performed the experiments, analyzed the data, contributed reagents/materials/analysis tools, wrote the paper, prepared figures and/or tables, reviewed drafts of the paper.

### DNA Deposition

The following information was supplied regarding the deposition of DNA sequences:
Sequences for isolates used in coculture assays are available in GenBank (KT598279, KT598280, KT598281, KT598282, KT598283, KT598284, KT598285 and KT598286).

## Data Availability

Raw sequence reads are available through the DDBJ (DNA Data Bank of Japan): PRJDB4237.

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
