# Peer review of "No apparent correlation between honey bee forager gut microbiota and honey production"

_PeerJ, doi:10.7717/peerj.1329_

## Round 0.1 · original submission · Major Revisions

The manuscript needs major revisions as both reviews suggest the same.

The fist reviewer is not an expert on bees but is qualified in bioinformatics, she felt it is an important contribution, where the authors make well addressed comparisons between productive and unproductive honey bee colonies. However, I think it can be improved in some aspects. First of all, it is not clear why they only sampled the gut microbiome, given that they have worked with the whole bee microbiome in previous papers, and that honey production has nothing to do with the gut. In this regard, I think it could be addressed in the title, but definitively in the abstract. In the introduction they should explain why the authors limit the work to the gut microbiome.

On the other hand, the second reviewer is an expert in bees and considered that the title is misleading. It suggests there is a correlation of gut bacteria with colony productivity but there is not. Because the culturing results are non-applicable to the rest of the findings, or what actually occurs in the gut, the manuscript amounts to a collection of 16S amplicons based on pollen foragers. There is another recent publication based entirely on 16S amplicons from pollen foragers, but it was completely ignored in the discussion. There are huge differences between these results, calling for a discussion of fundamental approaches that underlie methods and taxonomy. As there is no difference between treatments, this might be a discussion topic, adding a comparative aspect. It demonstrates that the authors do not want to consider the taxonomic methods and conclusions of others, but would rather spend time constructing elaborate interaction networks on taxonomy that does not agree with the rest of the field. The biggest red flag is the complete lack of Lactobacillus Firm5, known to dominate the guts of all worker bees including foragers. This result needs to be discussed up front. The one publication where Gammaproteobacteria was dominant in the honey bee gut resulted from a DNA isolation that lacked lysozyme and bead beating (Cox-Foster et al. 2006), suggesting methodological flaws, or some step in DNA isolation or amplification, has skewed these findings as well.

Reviewer 1 ·

Basic reporting

I feel it is an important contribution, where the authors make well addressed comparisons between productive and unproductive honey bee colonies. However, I think it can be improved in some aspects. First of all, it is not clear why they only sampled the gut microbiome, given that they have worked with the whole bee microbiome in previous papers, and that honey production has nothing to do with the gut. In this regard, I think it could be addressed in the title, but definitively in the abstract. In the introduction they should explain why the authors limit the work to the gut microbiome.

Regarding the references...is "NRDC" a valid reference?
The references to the figures are wrong, check the order and change it accordingly

Line 169: ...large number from THESE colonies... Which? You're starting a new section... The sampled colonies, I get it, but the last thing you were talking about is the co-cultures....

Figure 4 (the network). Gammaproteo appear as a hub, but I don't find this peculiar, since in most microbial communities this group tends to be a dominant taxa, specially in a gut community. I don't see the relevance of this figure at all, specially because you have two better ones showing some other correlations, and you could not find the same antagonistic interactions in the co-cultures. In any case, I think it would be more useful to have two networks, from the two different kind of colonies and see how the presence or absence of certain bacterial groups could alter the interactions between them.

Experimental design

I would strongly suggest the authors use more popular methods to determine if the microbial communities are different, such as Unifrac and PCA (if appropriate). The importance of using popular methods is that people can reproduce your results, besides both are well known methods that work perfect for microbial community comparisons.

The text is contradictory and sometimes the microbial populations of productive and unproductive colonies are significantly different (line 280) whereas in other sections they are not (line 290). Shouldn't the heading be "Microbial communities DOES NOT correlate with honey bee colony productivity"??

Regarding the co-culture analysis, I think it must be specified how the selection of strains was done and based on what criteria. This is a bit weird, given the fact that are not the bacteria isolated from the sampled bees. To know how the selection was made is important, because with your sequences you cannot get to the family level, so, how can you tell you made a correct choice? I can understand why this was done in this way, but it must be specified that is an approximation to try to explain the negative correlations found in the sequences. Additionally, I would have preferred to see this in plates and not in liquid media, since in plates you can see inhibition hales or overlapping colonies. Could it be possible to add this?
The phylogeny (Figure) is not referenced in the paragraph (line 160-161)

In line 167, "host many bacterial genera". Many as compared to what? Cheeses have, for example, 24 genera and are considered "low diverse"

Validity of the findings

Is it really that the productivity is associated with increased bacterial diversity?? And increased by just two taxa?? That's not much of an increase.... You state this in the abstract, but after reading the paper, I wasn't able to tell if an increase in the diversity (?) works better or worse for honey production. The presence of these two bacterial families appear to be significant, but as you state (lines 211-212), this correlation is not a causation of productivity. I would be more careful with this final statement. There is nothing wrong in reporting that the gut microbiome of the foragers has nothing to do with productivity. I believe it is a valuable finding.

Line 299-307: I would add to this discussion what would happen not only if you sampled different individuals, or see the transcriptome, but also if you looked at different microbiomes... the bacteria in the legs? exoskeleton? tongue? inside the comb?

In general, the results are described a bit chaotically. It is hard to follow all the correlations and where these are relevant. I would either try to simplify it and/or improve the wrap-up part in the discussion.

Reviewer 2 ·

Basic reporting

The title is misleading. It suggests there is a correlation of gut bacteria with colony productivity but there is not. Because the culturing results are non-applicable to the rest of the findings, or what actually occurs in the gut, the manuscript amounts to a collection of 16S amplicons based on pollen foragers. There is another recent publication based entirely on 16S amplicons from pollen foragers, but it was completely ignored in the discussion. There are huge differences between these results, calling for a discussion of fundamental approaches that underlie methods and taxonomy. As there is no difference between treatments, this might be a discussion topic, adding a comparative aspect. It demonstrates that the authors do not want to consider the taxonomic methods and conclusions of others, but would rather spend time constructing elaborate interaction networks on taxonomy that does not agree with the rest of the field. The biggest red flag is the complete lack of Lactobacillus Firm5, known to dominate the guts of all worker bees including foragers. This result needs to be discussed up front. The one publication where Gammaproteobacteria was dominant in the honey bee gut resulted from a DNA isolation that lacked lysozyme and bead beating (Cox-Foster et al. 2006), suggesting methodological flaws, or some step in DNA isolation or amplification, has skewed these findings as well.
I am confused as to why the co-culture assays are even presented as part of this manuscript. Concerning this data set, it has been demonstrated that the gut bacteria exist as a biofilm, having an intimate interaction with the epithelial lining of the ileum and rectum. The bacteria are co-evolved in their interaction with the host and with one another. Predictions based on what is known of biofilm communities would reflect limited resource space. However, the authors provide growth experiments that used nutrient rich broth and do not force intimate interactions like those that actually occur in a gut environment, but allow unrestricted growth where neither space nor nutrients are limiting the population. Perhaps a bigger flaw, the authors do not demonstrate that the bacteria characterized by amplicon sequencing are the bacteria they used in experimental assays. The authors are reasoning from a false premise, that correlations seen in vivo should also be evident in vitro. A critical part of this argument lies in the similarity of their in vitro environment to the actual environment occupied by the bacteria. There is the additional problem that the gram positive species occupy different gut space than do the gram negative species, so competition is unexpected.
The authors advise others to exercise caution when comparing culture assays to relative amplicon abundance. Maybe some caution should be exercised in the generation of in vitro assays such that they mimic real conditions, and use the actual gut strains found in these environments. It is not clear where the authors retrieved their co-cultured strains, or why they think those particular strains are the bacteria that should be tested against one another. It is highly likely that co-culture result are meaningless in the gut context.
Figure 1 state the non-significance in the figure caption.
Figure 4. is difficult to read, even more difficult to extract a message. Are the authors trying to show that like avery community there are positive and negative interactions? There are always many correlations in amplicon data, but largely overlooked in this analysis, the authors need to account for compositional constraints because the assumption of independence is violated.

Experimental design

The title is misleading. It suggests there is a correlation of gut bacteria with colony productivity but there is not. Because the culturing results are non-applicable to the rest of the findings, or what actually occurs in the gut, the manuscript amounts to a collection of 16S amplicons based on pollen foragers. There is another recent publication based entirely on 16S amplicons from pollen foragers, but it was completely ignored in the discussion. There are huge differences between these results, calling for a discussion of fundamental approaches that underlie methods and taxonomy. As there is no difference between treatments, this might be a discussion topic, adding a comparative aspect. It demonstrates that the authors do not want to consider the taxonomic methods and conclusions of others, but would rather spend time constructing elaborate interaction networks on taxonomy that does not agree with the rest of the field. The biggest red flag is the complete lack of Lactobacillus Firm5, known to dominate the guts of all worker bees including foragers. This result needs to be discussed up front. The one publication where Gammaproteobacteria was dominant in the honey bee gut resulted from a DNA isolation that lacked lysozyme and bead beating (Cox-Foster et al. 2006), suggesting methodological flaws, or some step in DNA isolation or amplification, has skewed these findings as well.
I am confused as to why the co-culture assays are even presented as part of this manuscript. Concerning this data set, it has been demonstrated that the gut bacteria exist as a biofilm, having an intimate interaction with the epithelial lining of the ileum and rectum. The bacteria are co-evolved in their interaction with the host and with one another. Predictions based on what is known of biofilm communities would reflect limited resource space. However, the authors provide growth experiments that used nutrient rich broth and do not force intimate interactions like those that actually occur in a gut environment, but allow unrestricted growth where neither space nor nutrients are limiting the population. Perhaps a bigger flaw, the authors do not demonstrate that the bacteria characterized by amplicon sequencing are the bacteria they used in experimental assays. The authors are reasoning from a false premise, that correlations seen in vivo should also be evident in vitro. A critical part of this argument lies in the similarity of their in vitro environment to the actual environment occupied by the bacteria. There is the additional problem that the gram positive species occupy different gut space than do the gram negative species, so competition is unexpected.
The authors advise others to exercise caution when comparing culture assays to relative amplicon abundance. Maybe some caution should be exercised in the generation of in vitro assays such that they mimic real conditions, and use the actual gut strains found in these environments. It is not clear where the authors retrieved their co-cultured strains, or why they think those particular strains are the bacteria that should be tested against one another. It is highly likely that co-culture result are meaningless in the gut context.
Figure 1 state the non-significance in the figure caption.
Figure 4. is difficult to read, even more difficult to extract a message. Are the authors trying to show that like avery community there are positive and negative interactions? There are always many correlations in amplicon data, but largely overlooked in this analysis, the authors need to account for compositional constraints because the assumption of independence is violated.

Validity of the findings

The title is misleading. It suggests there is a correlation of gut bacteria with colony productivity but there is not. Because the culturing results are non-applicable to the rest of the findings, or what actually occurs in the gut, the manuscript amounts to a collection of 16S amplicons based on pollen foragers. There is another recent publication based entirely on 16S amplicons from pollen foragers, but it was completely ignored in the discussion. There are huge differences between these results, calling for a discussion of fundamental approaches that underlie methods and taxonomy. As there is no difference between treatments, this might be a discussion topic, adding a comparative aspect. It demonstrates that the authors do not want to consider the taxonomic methods and conclusions of others, but would rather spend time constructing elaborate interaction networks on taxonomy that does not agree with the rest of the field. The biggest red flag is the complete lack of Lactobacillus Firm5, known to dominate the guts of all worker bees including foragers. This result needs to be discussed up front. The one publication where Gammaproteobacteria was dominant in the honey bee gut resulted from a DNA isolation that lacked lysozyme and bead beating (Cox-Foster et al. 2006), suggesting methodological flaws, or some step in DNA isolation or amplification, has skewed these findings as well.
I am confused as to why the co-culture assays are even presented as part of this manuscript. Concerning this data set, it has been demonstrated that the gut bacteria exist as a biofilm, having an intimate interaction with the epithelial lining of the ileum and rectum. The bacteria are co-evolved in their interaction with the host and with one another. Predictions based on what is known of biofilm communities would reflect limited resource space. However, the authors provide growth experiments that used nutrient rich broth and do not force intimate interactions like those that actually occur in a gut environment, but allow unrestricted growth where neither space nor nutrients are limiting the population. Perhaps a bigger flaw, the authors do not demonstrate that the bacteria characterized by amplicon sequencing are the bacteria they used in experimental assays. The authors are reasoning from a false premise, that correlations seen in vivo should also be evident in vitro. A critical part of this argument lies in the similarity of their in vitro environment to the actual environment occupied by the bacteria. There is the additional problem that the gram positive species occupy different gut space than do the gram negative species, so competition is unexpected.
The authors advise others to exercise caution when comparing culture assays to relative amplicon abundance. Maybe some caution should be exercised in the generation of in vitro assays such that they mimic real conditions, and use the actual gut strains found in these environments. It is not clear where the authors retrieved their co-cultured strains, or why they think those particular strains are the bacteria that should be tested against one another. It is highly likely that co-culture result are meaningless in the gut context.
Figure 1 state the non-significance in the figure caption.
Figure 4. is difficult to read, even more difficult to extract a message. Are the authors trying to show that like avery community there are positive and negative interactions? There are always many correlations in amplicon data, but largely overlooked in this analysis, the authors need to account for compositional constraints because the assumption of independence is violated.

Additional comments

The title is misleading. It suggests there is a correlation of gut bacteria with colony productivity but there is not. Because the culturing results are non-applicable to the rest of the findings, or what actually occurs in the gut, the manuscript amounts to a collection of 16S amplicons based on pollen foragers. There is another recent publication based entirely on 16S amplicons from pollen foragers, but it was completely ignored in the discussion. There are huge differences between these results, calling for a discussion of fundamental approaches that underlie methods and taxonomy. As there is no difference between treatments, this might be a discussion topic, adding a comparative aspect. It demonstrates that the authors do not want to consider the taxonomic methods and conclusions of others, but would rather spend time constructing elaborate interaction networks on taxonomy that does not agree with the rest of the field. The biggest red flag is the complete lack of Lactobacillus Firm5, known to dominate the guts of all worker bees including foragers. This result needs to be discussed up front. The one publication where Gammaproteobacteria was dominant in the honey bee gut resulted from a DNA isolation that lacked lysozyme and bead beating (Cox-Foster et al. 2006), suggesting methodological flaws, or some step in DNA isolation or amplification, has skewed these findings as well.
I am confused as to why the co-culture assays are even presented as part of this manuscript. Concerning this data set, it has been demonstrated that the gut bacteria exist as a biofilm, having an intimate interaction with the epithelial lining of the ileum and rectum. The bacteria are co-evolved in their interaction with the host and with one another. Predictions based on what is known of biofilm communities would reflect limited resource space. However, the authors provide growth experiments that used nutrient rich broth and do not force intimate interactions like those that actually occur in a gut environment, but allow unrestricted growth where neither space nor nutrients are limiting the population. Perhaps a bigger flaw, the authors do not demonstrate that the bacteria characterized by amplicon sequencing are the bacteria they used in experimental assays. The authors are reasoning from a false premise, that correlations seen in vivo should also be evident in vitro. A critical part of this argument lies in the similarity of their in vitro environment to the actual environment occupied by the bacteria. There is the additional problem that the gram positive species occupy different gut space than do the gram negative species, so competition is unexpected.
The authors advise others to exercise caution when comparing culture assays to relative amplicon abundance. Maybe some caution should be exercised in the generation of in vitro assays such that they mimic real conditions, and use the actual gut strains found in these environments. It is not clear where the authors retrieved their co-cultured strains, or why they think those particular strains are the bacteria that should be tested against one another. It is highly likely that co-culture result are meaningless in the gut context.
Figure 1 state the non-significance in the figure caption.
Figure 4. is difficult to read, even more difficult to extract a message. Are the authors trying to show that like avery community there are positive and negative interactions? There are always many correlations in amplicon data, but largely overlooked in this analysis, the authors need to account for compositional constraints because the assumption of independence is violated.

---

## Round 0.2 · accepted · Accept

I have read the new version of the manuscript and the rebuttal letter and I feel that all the observations made by the reviewers have been acceptably addressed.